EMBO
Molecular Medicine

# Chronic miR-29 antagonism promotes favorable plaque remodeling in atherosclerotic mice

Victoria Ulrich[1,2], Noemi Rotllan[1,3], Elisa Araldi[1,3], Amelia Luciano[1,2], Philipp Skroblin[5], Mélanie Abonnenc[5], Paola Perrotta[1,2], Xiaoke Yin[5], Ashley Bauer[1,6], Kristen L Leslie[1,6], Pei Zhang[1,2], Binod Aryal[1,3], Rusty L Montgomery[4], Thomas Thum[7,8], Kathleen Martin[1,6], Yajaira Suarez[1,3], Manuel Mayr[5], Carlos Fernandez-Hernando[1,3] & William C Sessa[1,2,*]

## Abstract

Abnormal remodeling of atherosclerotic plaques can lead to rupture, acute myocardial infarction, and death. Enhancement of plaque extracellular matrix (ECM) may improve plaque morphology and stabilize lesions. Here, we demonstrate that chronic administration of LNA-miR-29 into an atherosclerotic mouse model improves indices of plaque morphology. This occurs due to upregulation of miR-29 target genes of the ECM (*col1A and col3A*) resulting in reduced lesion size, enhanced fibrous cap thickness, and reduced necrotic zones. Sustained LNA-miR-29 treatment did not affect circulating lipids, blood chemistry, or ECM of solid organs including liver, lung, kidney, spleen, or heart. Collectively, these data support the idea that antagonizing miR-29 may promote beneficial plaque remodeling as an independent approach to stabilize vulnerable atherosclerotic lesions.

**Keywords** atherosclerosis; LNA; miR-29; plaque; stability
**Subject Categories** Cardiovascular System; Immunology

## Introduction

Cardiovascular disease (CVD) is the leading cause of death worldwide. The development of atherosclerosis is triggered by elevation of plasma low-density lipoprotein (LDL) cholesterol initiating early endothelial dysfunction, inducing innate immune responses and the development of lipid-laden plaques. The inflammatory response within the arterial wall due to the presence of immune cells, accumulated cholesterol crystals, and necrotic zones with the atheroma can lead to remodeling of the extracellular matrix (ECM) of the vessel and lesional fibrous caps (Small *et al*, 1984; Schwenke & Carew, 1989; Gimbrone *et al*, 1995; Arroyo & Lee, 1999; Abela & Aziz, 2005; Heidenreich *et al*, 2011). In later stages of atherosclerosis, plaques can mature and often become unstable, rupture, and cause acute ischemic events. The thrombotic events resulting from severe coronary and cerebrovascular atherosclerosis cause more deaths in developed countries than cancer and injuries, and the prevalence in developing countries is increasing with advances in health care and longer life spans (Kolodgie *et al*, 2001; Heidenreich *et al*, 2011; ISTH Steering Committee for World Thrombosis Day, 2014; Moran *et al*, 2014).

Recent data have shown that targeting ECM components in the vasculature may influence the progression of vascular diseases, indicating a potential new approach to remodel atherosclerotic lesions (Maegdefessel *et al*, 2012; Zhang *et al*, 2012a, 2013; Abonnenc *et al*, 2013; Melo *et al*, 2014). MicroRNAs (miRNAs) are evolutionarily conserved, single-stranded RNAs that have emerged as critical regulators of gene expression. miRNAs reduce the levels of expression by antisense or translational repressive mechanisms (Ambros, 2004; Filipowicz *et al*, 2008; Bartel, 2009). In the last decade, numerous studies have identified miRNAs that can regulate the ECM, and miR-29 is one of the most well-characterized miRNAs that targets several ECM mRNAs.

The miR-29 family consists of three largely homologous members, miR-29a, miR-29b, and miR-29c, and is highly expressed in fibroblasts and vascular smooth muscle cells (VSMC). While one miRNA often affects many different pathways within the cell or neighboring cells, the miR-29 family largely represses transcripts of at least 16 ECM components (van Rooij *et al*, 2008; Sengupta *et al*, 2008; Li *et al*, 2009; Liu *et al*, 2010; Chen *et al*, 2011). Using

1  Vascular Biology and Therapeutics Program, Yale University School of Medicine, New Haven, CT, USA
2  Department of Pharmacology, School of Medicine, Yale University, New Haven, CT, USA
3  Integrative Cell Signaling and Neurobiology of Metabolism Program, Section of Comparative Medicine and Department of Pathology, School of Medicine, Yale University, New Haven, CT, USA
4  miRagen Therapeutics, Boulder, CO, USA
5  King's British Heart Foundation Centre, King's College London, London, UK
6  Department of Cardiology, School of Medicine, Yale University, New Haven, CT, USA
7  Institute of Molecular and Translational Therapeutic Strategies, Hannover Medical School, Hannover, Germany
8  National Heart and Lung Institute, Imperial College London, London, UK
   *Corresponding author. Tel: +1 203 737 2291; Fax: +1 203 737 2290; E-mail: william.sessa@yale.edu

anti-miRNAs (either cholesterol-modified or locked nucleic acid analogs to base pair with endogenous miRNAs) to antagonize endogenous miRNA, we and others have successfully targeted miR-29 *in vivo* and *in vitro*, and have consistently demonstrated increased ECM components to restore vessel dynamics without obvious off-target effects. Interestingly, while several components of the ECM decrease in CVD, miR-29 expression increases in the vasculature (van Rooij *et al*, 2008; Boon *et al*, 2011; Takahashi *et al*, 2012; Zhang *et al*, 2012a,b; Yang *et al*, 2013; James *et al*, 2014; Han *et al*, 2015). Thus, we investigated whether the modulation of miR-29 and its putative gene targets affects atheroma formation and composition, and might offer a new approach to promote favorable plaque remodeling.

# Results

### LNA-miR-29 decreases atherosclerotic lesion size

Given the importance of miR-29 in vascular homeostasis and the posttranscriptional regulation of many extracellular matrix proteins including elastin and collagen, and the evidence that antagonizing miR-29 can rescue genetic deficiency states in cells and influence aneurysm expansion (van Rooij *et al*, 2008; Sengupta *et al*, 2008; Li *et al*, 2009; Liu *et al*, 2010; Boon *et al*, 2011; Chen *et al*, 2011; Maegdefessel *et al*, 2012; Merk *et al*, 2012; Zhang *et al*, 2012a; Ekman *et al*, 2013), we chose to investigate whether targeting of miR-29 *in vivo* would alter atherogenesis and plaque composition. To this end, 8-week-old $ApoE^{-/-}$ mice fed a Western diet (WD) were subcutaneously administered saline, control, or a miR-29 inhibitor as a locked nucleic acid (called LNA-control and LNA-miR-29, administered at 4 mg/kg, respectively) biweekly for 14 weeks (Fig 1A). The LNA-miR-29 belongs to a class of oligonucleotides with classical LNA-containing oligonucleotide pharmacokinetic profiles (Elmen *et al*, 2008; Hullinger *et al*, 2012). In brief, following subcutaneous administration, plasma concentrations for these anti-miRs typically achieve peak concentrations between 30 min and 1 h after administration. Plasma clearance is biphasic with a short, initial distribution phase, followed by a longer elimination phase. Oligonucleotide accumulation is highest in the kidney and liver, with significant accumulation also observed in spleen, bone marrow, and distal skin (away from the injection site). Terminal elimination half-lives are several weeks, ranging from roughly 3–6 weeks. Administration of these agents had no effect on total cholesterol, triglycerides, lipid profiles, and blood cell composition assayed at the time of harvest (Figs EV1A and EV2). However, LNA-miR-29 (4 mg/kg) effectively neutralized all miR-29 isoforms in carotid arteries (Fig EV1B) and reduced lesion size in aortic roots and brachiocephalic arteries (BC; Fig 1B and C).

### LNA-miR-29 promotes beneficial plaque remodeling consistent with stable lesions

Luminal fibrous caps on remodeled atherosclerotic plaques are composed mainly of VSMC and a collagenous ECM, and there is evidence that thicker fibrous caps are less prone to rupture (Davies *et al*, 1993; Galis *et al*, 1994; Nikkari *et al*, 1995; Aikawa *et al*, 1998; Sukhova *et al*, 1999; Fernandez-Hernando *et al*, 2009; Maurer *et al*, 2010). Treatment with LNA-miR-29 visibly increased fibrous cap thickness in both roots and BC (Fig 2A, quantified in Fig 2B) and levels of immunoreactive smooth muscle actin (SMA; Fig 2C and D) but not the abundance of a macrophage marker (CD68), proliferating cells (Ki-67), or an apoptotic marker (TUNEL; Figs EV3 and EV4), indicating a relatively quiescent population of VSMC in LNA-miR-29-treated group. Since increased necrotic areas can destabilize atherosclerotic plaques (Tracy, 1984), necrotic areas within the plaques were measured. Compared to LNA-control-injected mice, the LNA-miR-29 group exhibited significantly smaller necrotic areas (Fig 2E and F).

### LNA-miR-29 induces intra-plaque collagen gene expression but not systemic fibrosis

Plaque stability is directly proportional to subluminal collagen content (Aikawa *et al*, 1998), with COL1A and COL3A1 subtypes being the most prevalent in atherosclerotic plaques; both are well-described targets of miR-29 (Shekhonin *et al*, 1987; Sengupta *et al*, 2008; Liu *et al*, 2010). To assess COL expression, $ApoE^{-/-}$ mice fed a WD were treated as above with saline, control LNA, or LNA-miR-29, and common carotid arteries (CCA) were excised and dissected into visible plaque versus visible non-plaque areas to evaluate any changes in COL gene expression within plaques by qPCR. *COL1A1* and *COL3A1* mRNA expression levels were increased in plaque (P) versus non-plaque (NP) tissue (Fig 3A). Collagen protein deposition within root and BC plaques was then analyzed utilizing picrosirius red staining and polarized light microscopy in BC lesions (Fig 3B, quantified in 3C) and aortic root (Fig 3D, quantified in 3E). Intraplaque collagen levels were significantly higher in LNA-miR-29 tissues compared to saline and LNA-control tissues in root lesions and in the BC lesions. However, LNA-miR-29 did not cause systemic fibrosis in several non-vascular organs including liver, lung, kidney, spleen, and heart (Fig EV5A, quantified in Fig EV5B) (Boon *et al*, 2011).

### Modulation of miR-29 in primary VSMC regulates secreted ECM proteins

Since LNA-miR-29 increased SMA-positive cells in fibrous caps and reduced necrotic zones and VSMC contribute to collagen deposition within plaques (Heinrich *et al*, 1995), the miR-29-regulated proteome was studied in VSMC *in vitro*. Murine VSMC were transfected with miR-29 pre-miR or anti-miR-29, and the secreted protein of miR-29 targets identified via quantitative proteomics. The proteins are ordered from the smallest to largest false discovery rate (FDR) with significant differential expression set at an FDR 5% ($P < 0.05$) utilizing a model based on a hierarchical Bayesian estimation of generalized linear mixed-effects model (Choi *et al*, 2008). This analysis identified several collagen subtypes (COL1A1, COL1A2, COL3A1, COL4A1, COL5A2), fibronectin 1 (FBN1), and MMP2 as the most differentially secreted, miR-29 target proteins repressed by miR-29. Conversely, collagenous protein levels were upregulated by suppressing miR-29. Using this analysis, COL3A1 was the most highly regulated protein following both mimic and anti-miR treatment (Fig 4A; Appendix Table S1A and B). Since COL3A is an important collagen subtype in plaques (Shekhonin

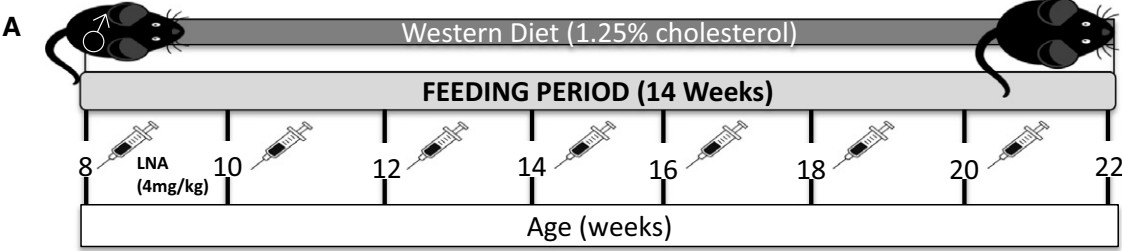

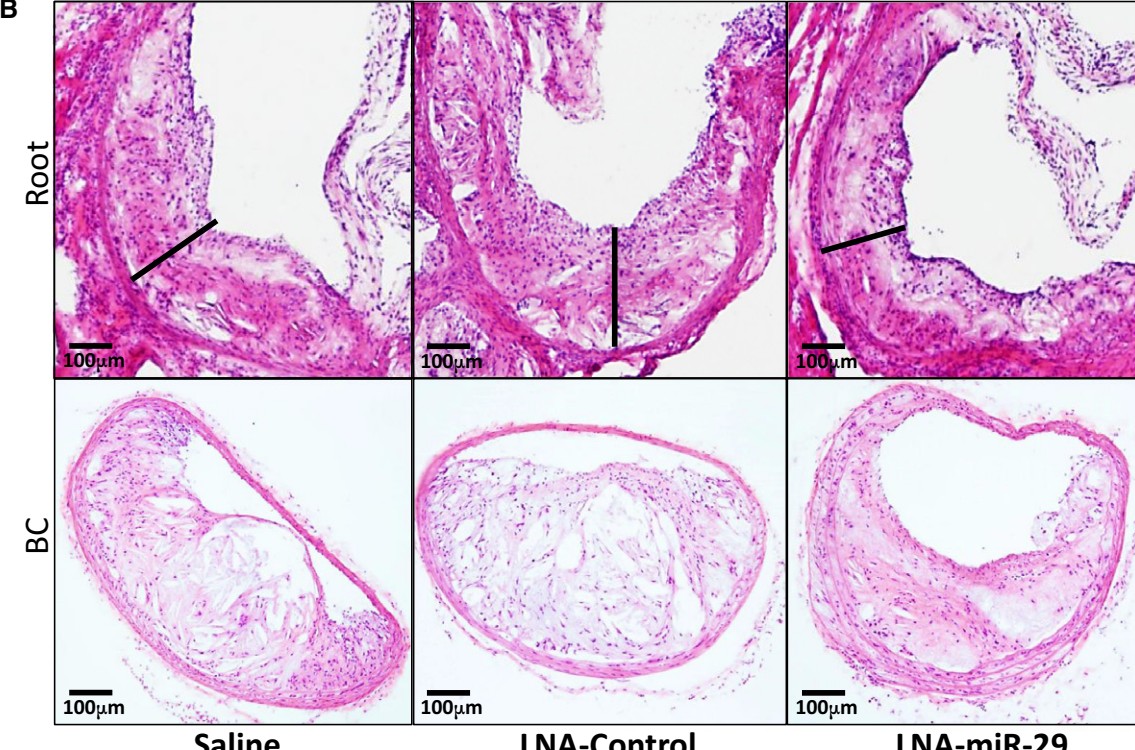

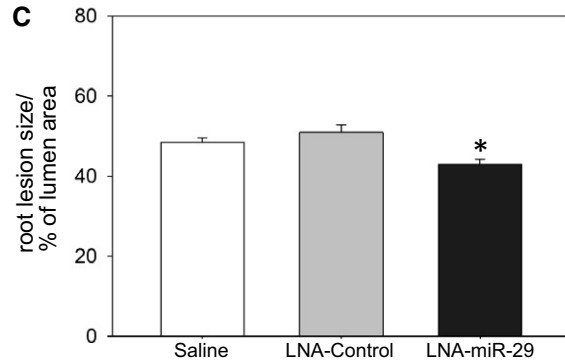

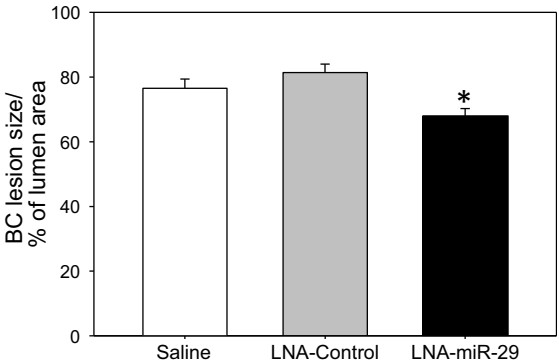

**Figure 1.  LNA-miR-29 reduces atherosclerosis lesion size.**

A   Schematic representation of experimental mouse design. Briefly, 8-week-old male *ApoE*$^{-/-}$ mice on Western diet were injected with equal volume of saline, or
     4 mg/kg of LNA-control, or LNA-miR-29 biweekly for 14 weeks.

B   H&E staining of representative atherosclerotic lesions in aortic root (upper panel: 4× magnification) and brachiocephalic arteries (lower panel: 10× magnification) at
     14 week harvest. Lesion width of roots highlighted with black bars. Scale bars, 100 μm.

C   Quantification of root lesion size (left panel, *P = 0.002), and of brachiocephalic lesion size (right panel, *P = 0.001), both quantified as percent of total lumen area.
     Data are averaged lesion percentages from 4- to 6-μm serial sections from each animal, *n* = 9, 13, 15. All of the data represent mean ± SEM, *P < 0.05 versus LNA-
     control tissue. Level of significance was determined using one-way ANOVA with Bonferroni's post-test.

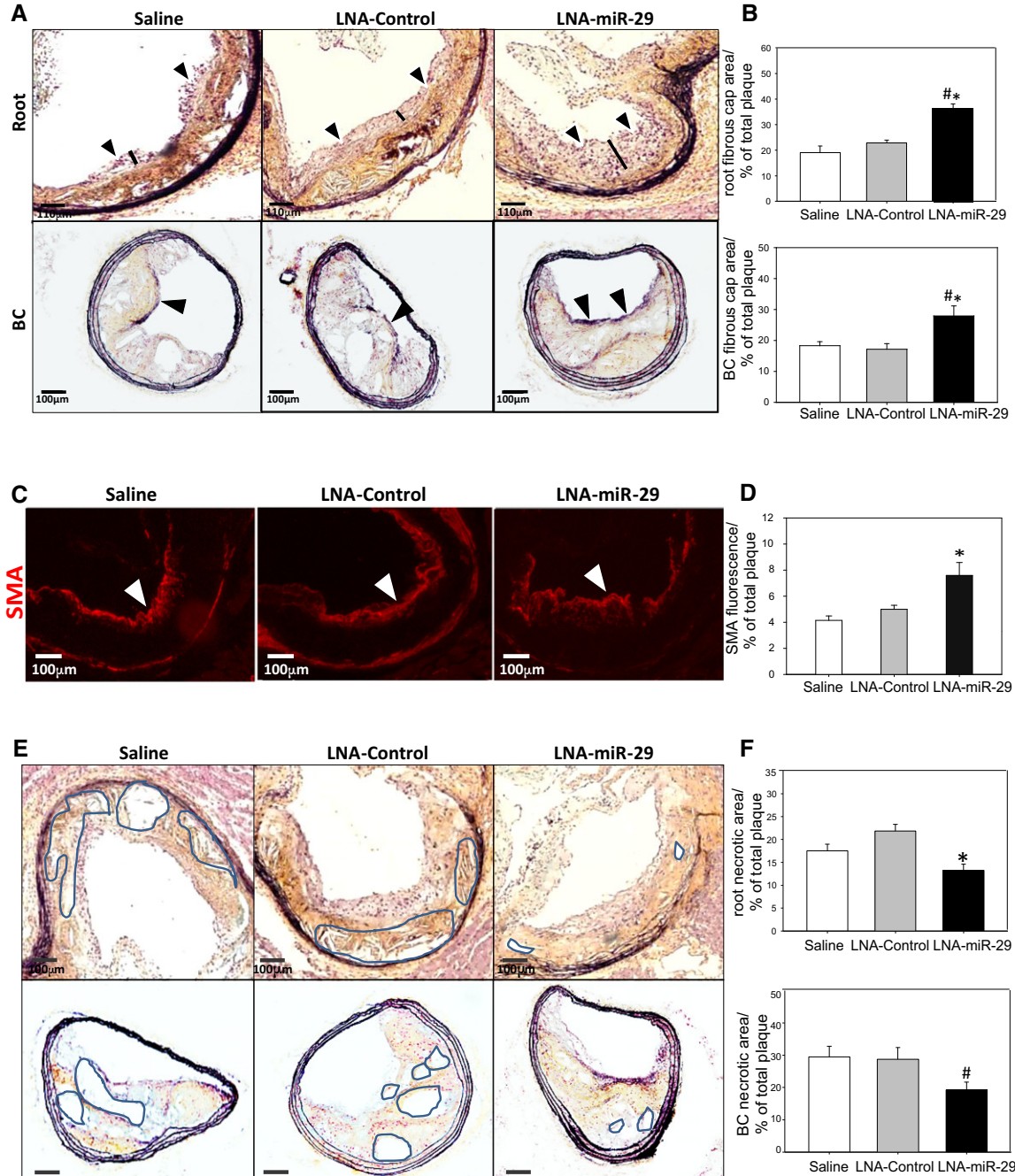

**Figure 2. LNA-miR-29 increases fibrous cap thickness and SMA staining and reduces necrotic zones in lesions.**

A Movat pentachrome staining of aortic root (upper panel: 4× magnification) and brachiocephalic artery fibrous caps (lower panel: 10× magnification) at 14 week harvest, caps denoted with arrowheads and black bars (roots only). Scale bars, 110 μm (upper row) and 100 μm (lower row).

B Quantification of aortic root (upper panel, *P = 0.005, #P < 0.001) and brachiocephalic artery (lower panel *P = 0.011, #P < 0.024) fibrous caps, graphed as percent of total plaque area.

C, D Representative immunofluorescence (IF) staining of smooth muscle actin (red), denoted by white arrowheads, in aortic root lesions shown at 10× magnification. IF images in (C) are representative of data quantified in (D) calculated from 3- to 6-μm serial sections from animals in independent cohorts. Data represent mean ± SEM, n = 9, 13, 15 for the three groups. Level of significance was determined using Kruskal–Wallis one-way ANOVA on ranks, *P = 0.002 (D). Scale bars, 100 μm.

E Movat pentachrome staining of aortic root (upper panel: 4× magnification, *P < 0.001) and brachiocephalic artery necrotic areas (lower panel: 10× magnification) at 14 week harvest, representative necrotic zones denoted by highlighted areas. Scale bars, 100 μm.

F Quantification of aortic root (left panel, *P < 0.001) and brachiocephalic artery (right panel, #P = 0.044) necrotic areas, graphed as percent of total plaque area.

Data information: Data in (B, F) are averaged percentages per total plaque area from 4- to 6-μm serial sections from each animal. All of the data represent mean ± SEM, *P < 0.05 versus LNA-control tissue, # represents P < 0.05 versus saline tissue, n = 9, 13, 15. Level of significance was determined using one-way ANOVA with Bonferroni's post-test.

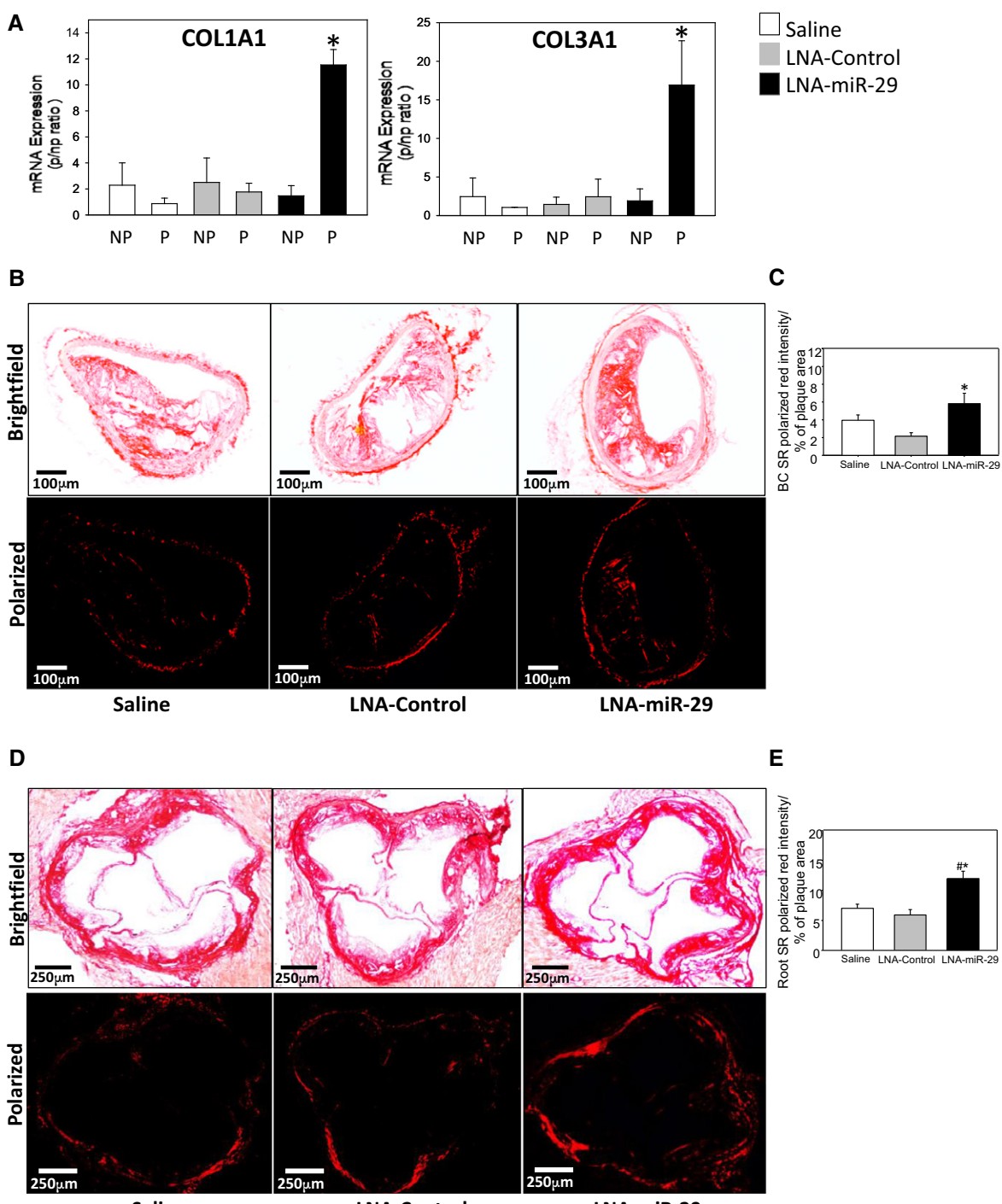

**Figure 3.  LNA-miR-29 increases collagen in murine plaques.**

*ApoE*$^{-/-}$ common carotid arteries (CCA) were dissected into plaque and adjacent non-plaque areas, then RNA isolated.

A      Graphed qPCR quantification of collagen 1A1 (left panel) and collagen 3A1 (right panel) mRNA expression in plaque (P) and non-plaque (NP) areas of *ApoE*$^{-/-}$ carotid arteries. Three assays were carried out, and the data represent mRNA expression normalized to NP, and mean ± SEM, *n* = 4, 3, 4 of mice/group. *P < 0.001 compared with LNA-miR-29 non-plaque region.

B–E    Picrosirius red staining of collagen in BCs (10× magnification; scale bars, 100 μm) (B); and aortic roots (4× magnification; scale bars, 250 μm) (D); brightfield (upper panel) and polarized light (lower panel) imaging shown. Collagen quantification in BCs (C, *P = 0.016 versus LNA-control tissue) and roots (E, *P = 0.003 versus LNA-control tissue, #P = 0.005 versus saline tissue) from polarized light images expressed as a percentage of plaque area. Data represent mean ± SEM, n = 9, 13, 15.

Data information: Data in (C, E) are averaged percentages per total plaque area from 4- to 6-μm serial sections from each animal. Level of significance (A, C, E) was determined using one-way ANOVA with Bonferroni's post-test.

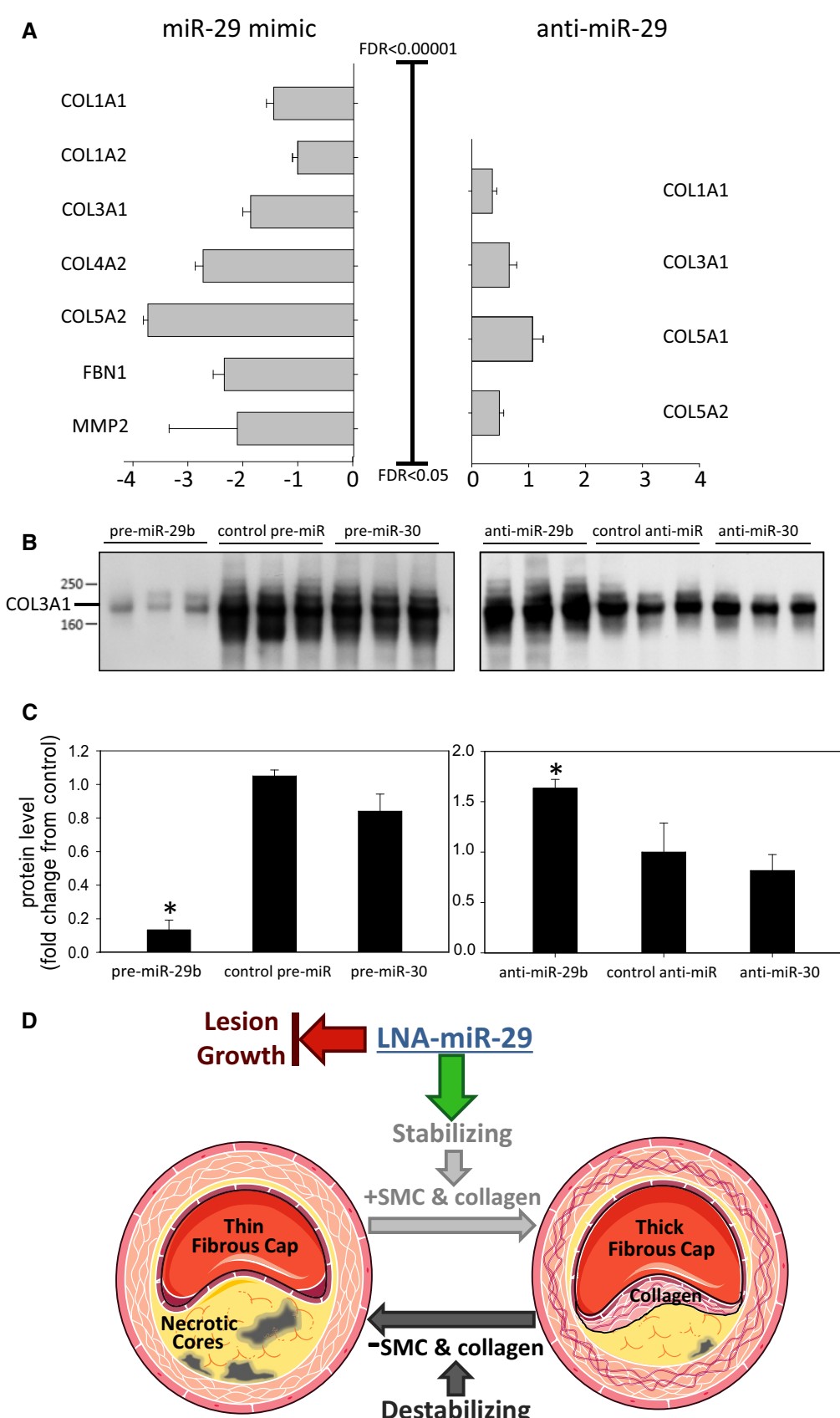

Figure 4.

**Figure 4.  Modulation of miR-29 defines murine VSMC proteome and collagen targets.**
Proteome studies analyze the frequency distribution of protein changes after the transfection of primary murine aortic VSMC with pre-miR-29b, anti-miR-29b, pre-miR-30c, or anti-miR-30c. A log2 fold change of −1 or 1 was used as cutoff. Log2 (fold change) is calculated using the normalized spectral counts and averaged for the 3 biological replicates.

A   The log (fold change) for differentially secreted proteins following overexpression or inhibition of pre-miR-29b was analyzed, and significantly changed miR-29 confirmed target transcripts are shown. FDR indicates false discovery rate. The complete list of significant proteins is in Appendix Table S1A and B. All data represent mean ± SEM.

B   Thirty microliters of conditioned medium from VSMC transfected with pre-miR-29b and anti-miR-29b was separated by SDS–PAGE before validation of COL3A1 by immunoblotting.

C   Densitometry analysis of COL3A1 signal from blot in (B) (pre-miR, left *P < 0.001, anti-miR, right *P = 0.002). All data in (C) represent mean ± SEM, *P < 0.05 versus other groups in anti- and pre- groups. Level of significance was determined using one-way ANOVA with Bonferroni's post-test.

D   Working model of LNA-miR-29 effects on plaque composition as detailed in the data of this paper. Administration of LNA-miR-29 blocks lesion growth and increases intra-plaque collagen expression and fibrous caps, two indices characteristic of more stable plaques. This panel was produced using Servier Medical Art (www.servier.com).

*et al*, 1987), the specific levels of COL3A secretion were further validated by immunoblotting of conditioned medium (Fig 4B), and quantified (Fig 4C) in mimic versus anti-miR-treated cells. miR-30 (pre- and anti-miR-30) was used as an additional control for specificity. Collectively, these data demonstrate broad control of ECM protein levels by miR-29 in VSMC.

## Discussion

Here, we demonstrate that chronic administration of LNA-miR-29 into an atherosclerotic mouse model improves indices of plaque morphology. This occurs due to upregulation of miR-29 target genes of the ECM (*Col1A1* and *Col3A1*) resulting in reduced lesion size, enhanced fibrous cap thickness, and reduced necrotic zones (see Fig 4D for model). Sustained LNA-miR-29 treatment did not affect circulating lipids, blood chemistry, or ECM of solid organs including liver, lung, kidney, spleen, or heart (Figs EV1A, EV2, and EV5A and B). Collectively, these data support the idea that antagonizing miR-29 may promote beneficial plaque remodeling as a new approach to potentially stabilize vulnerable atherosclerotic lesions.

miR-29 uniquely targets several genes that regulate ECM production and may influence disease pathologies associated with abnormal ECM production. Previous work has shown that anti-miR-29 can upregulate elastin mRNA and protein levels in two diseases of elastin haploinsufficiency, Williams–Beuren syndrome (WBS) and supraventricular aortic stenosis (Zhang *et al*, 2012b). Chronic administration of LNA-miR-29 reduces aortic aneurysm formation in angiotensin II and Marfan syndrome models via upregulation of several ECM genes (Merk *et al*, 2012) providing *in vivo* evidence supporting the promise of miR-29 antagonism reducing the extent of vascular diseases. Our data show that chronic LNA-miR-29 treatment in a well-accepted mouse model of atherosclerosis increases indices of plaque stability, indicating a potential role for modulation of miR-29 to affect plaque size and composition. Indeed, most deaths from coronary artery disease are due to disruption of thinning fibrous caps overlying necrotic cores in plaques resulting in plaque thrombosis and embolism (Zhou *et al*, 1999; Kolodgie *et al*, 2001; ISTH Steering Committee for World Thrombosis Day, 2014). One caveat of our study is that it is well appreciated that murine models of atherosclerosis mimic certain aspects of human atheroma, but not plaque rupture. However, the reduction in plaque necrotic zones and fibrous cap thickening are consistent with the concept that LNA-miR-29 may reduce plaque instability in models that exhibit vulnerable plaque and its embolic complications.

Another consideration relevant to strategies aimed at antagonizing miR-29 function would be the potential for enhancing fibrosis in healthy tissues. However, there are data showing that miR-29 mimicry reduces pathological fibrosis in multiple solid organs with no observable effect of physiological turnover of ECM (Montgomery *et al*, 2014). In addition, previous work with LNA-miR-29 at five times the dose used in the present study (Boon *et al*, 2011) saw no systemic fibrosis in multiple tissues consistent with our data. Interestingly, in the present study LNA-miR-29 affected collagen content in plaques, but not in non-remodeling vessels, implying that LNA-miR-29 may exert its dominant effect in actively remodeling, diseased tissue expressing altered miR-29 levels. The mechanistic basis for why LNA-miR-29 only affects diseased tissue is not known.

The plaques of LNA-treated mice show increased SMA staining in the larger fibrous caps indicating a more quiescent VSMC population with increased collagen levels. Moreover, LNA-treated mice show less necrotic zones in plaques, but no differences in TUNEL staining across the groups. This could be due to the rapid kinetics of a low percentage of TUNEL-positive cells detected at single points when necrotic zones were established over the course of many weeks and/or beneficial remodeling of cellular constituents in the plaque including VSMC and macrophages. The improvement in plaque composition is unlikely related to net macrophage accumulation since CD68-positive macrophages were not different among the groups; however, individual subsets were not examined. Therefore, it is likely that the integrated response of LNA-treated mice could be the result of many factors, such as a slower plaque progression, more remodeling/regression, a smaller denser plaque due to less necrosis/more collagen content, or accelerated VSMC stability and quiescence. Collectively, LNA-miR-29 therapy reduces murine lesion size, and also improves indices of plaque stability through increased secretion of ECM and presents a therapeutic opportunity independent of lipid-lowering strategies for the treatment of atherosclerosis.

## Materials and Methods

### Experimental animals

Eight-week-old male *ApoE*$^{-/-}$ mice obtained from Jackson Laboratories (B6.129P2-Apoe tm1/Unc/J, #002052, Bar Harbor, ME) were fed a WD (D12108C, Research Diets, New Brunswick, NJ) containing 40% kcal from fat and 1.25% cholesterol, and kept under constant temperature and humidity in a 12-h controlled dark/light cycle. During WD feeding for 14 weeks, mice were injected

subcutaneously with sterile saline, or 4 mg/kg control, or microRNA-29-targeted locked nucleic acid (LNA-control, LNA-miR-29), respectively, biweekly. All of the experiments were approved by and comply with the Institutional Animal Care Use Committee of Yale University School of Medicine and ARRIVE animal guidelines. Treatment injections in mice were randomized such that each cage had several differentially treated animals within. Image data and samples were blinded by a third party prior to analysis.

## Lipids and lipoprotein profile

Mice were fasted for 12–14 h before blood samples were collected by cardiac puncture and plasma was separated by centrifugation and stored at −80°C. Total plasma cholesterol and triglycerides were enzymatically measured (Wako Pure Chemicals, Tokyo, Japan) according to the manufacturer's instructions. The lipid distributions in plasma lipoprotein fractions were assessed by fast-performed liquid chromatography gel filtration with 2 Superose 6 HR 10/30 columns (Pharmacia Biotech, Uppsala, Sweden).

## Blood cell analysis

Blood was collected by retro-orbital puncture in heparinized microhematocrit capillary tubes 4 days prior to harvest. Erythrocytes were lysed with ACK lysis buffer (155 mM ammonium chloride, 10 mM potassium bicarbonate, 0.01 mM EDTA, pH 7.4). WBC were resuspended in 3% FBS in PBS, blocked with 2 μg/ml of FcgRII/III, and then stained with a cocktail of antibodies. Monocytes were identified as CD115$^{hi}$ and subsets as Ly6-C$^{hi}$ and Ly6-C$^{lo}$; neutrophils were identified as CD115$^{lo}$Ly6-C$^{hi}$Ly6-G$^{hi}$. T cells were identified as CD4-high and CD8-high, respectively. Mature B cells were identified as B220-high, CD19-high, and IgM-high cells, while B cell progenitors were identified as CD19-high, B220-high, and IgM-low. Flow cytometry was performed using a BD LSRII and analyzed using FlowJo software (Tree Star). The following antibodies were used (all from BioLegend): FITC-Ly6-C (AL-21), PE-CD115 (AFS98), APC-Ly6-G (1A8), BV412-CD8 (53-6.7), APC-cy7-CD4 (RM4-5), PE-cy7-B220 (RA3-6B2), APC-CD19 (6D5), and PE-IgM (RMM-1); and all were used at a 1:500 dilution.

## Histology, immunohistochemistry, and morphometric analyses

Mouse hearts, brachiocephalic arteries (BC), lung, liver, kidney, and spleen were perfused with 10 ml of phosphate-buffered saline (PBS) at harvest and were submersed in 4% paraformaldehyde (PFA) overnight. After incubation in PFA, tissues were washed with PBS and left in PBS for 1 h. Next, tissues were put in 30% sucrose until the next day. Finally, tissues were embedded in OCT and frozen. Serial sections were cut at 6 μm thickness using a cryostat. Every third slide from the serial sections was stained with hematoxylin and eosin (H&E) or Movat pentachrome for quantification of lesion areas. Lesion size of each animal was obtained by averaging the lesion areas in four sections from the root and BC of the same mouse. Collagen content was assessed by picrosirius red staining of four sections from various tissues of the same mouse. CD68 staining was used as a macrophage marker using consecutive slides from serial sections (Serotec, MCA1957, 1:200) and was costained with monoclonal anti-actin α-smooth muscle-Cys antibody (SIGMA,

1:1,000) as an SMC marker. Nuclei were counterstained with DAPI for 10 min. The fibrous cap, necrotic core area, and macrophage content were measured as a percentage of the total plaque area from the four sections from the same mouse. Apoptotic cells in lesions were detected by TUNEL after proteinase K treatment, using the *In situ* Cell Death Detection kit, TMR red (Roche). Tricolor immunofluorescence staining was performed on cryosections by using different combinations of antibodies. For instance, in the tricolor labeling of TUNEL, CD68 or SMA, and DAPI, heart sections were first performed for TUNEL by following the manufacturer's instructions. Primary antibody for CD68 or SMA was then added and incubated for 1 h at RT. After PBS washes, secondary antibody was added for 45 min. Then, nuclei were counterstained with DAPI for 10 min. The data were expressed as the number of TUNEL-positive cells per CD68 or SMA plaque area. Proliferative cells in the lesion were detected by costained cryosections using Ki-67 (Abcam 66155, 1:100) and CD68 or SMA. Proliferating cells in the lesion were calculated as the number of positive Ki-67-labeled nuclei per CD68 or SMA plaque area. SMA or CD68 staining was quantified as % SMA or CD68 intensity staining/total plaque area. ImageJ software (NIH, USA) was used for all the quantifications.

## Isolation of plaque and non-plaque carotid arteries

At the time of harvest, common carotid arteries (CCA) from the inferior bifurcations at the aortic arch to the superior caudal bifurcation were exposed ventrally under dissecting microscope (Leica Microsystems, Buffalo Grove, IL), and cleaned of extraneous surrounding tissue. CCA were then wholly excised, and under higher magnification opaque vessel near bifurcations with visually observable plaques were dissected away from translucent, adjacent non-plaque vessel. Plaque and non-plaque vessel segments were then carefully places in separate 1.5-ml Eppendorf tubes, snap-frozen in liquid nitrogen, and stored at −80°C pending further processing as detailed below.

## RNA isolation and quantitative real-time PCR

Common carotid arteries were snap-frozen in liquid nitrogen, and then stored at −80°C pending further processing. Total RNA from arteries was isolated using the Bullet Blender Homogenizer (Next Advance) and the RNeasy Plus Mini Kit (Qiagen) according to the manufacturer's protocol. Following a brief incubation at ambient temperature, 140 μl of chloroform was added and the solution was mixed vigorously. The samples were then centrifuged at 5,000 *g* for 15 min at 4°C. The upper aqueous phase was carefully transferred to a new tube, and 1.5 volumes of ethanol was added. The samples were then applied directly to columns and washed according to the company's protocol. Total RNA was eluted in 25 μl of nuclease-free H$_2$O. RNA was quantified by NanoDrop (Agilent Technologies). A total of 1 μg of total RNA was reverse-transcribed using the iScript RT Supermix (Bio-Rad), following the manufacturer's protocol. Quantitative real-time PCR was performed in triplicate using iQ SYBR Green Supermix (Bio-Rad) on a Real-Time Detection System (Eppendorf). The mRNA level was normalized to GAPDH (glyceraldehyde-3-phosphate dehydrogenase) and miRNA normalized to RNU6 as a housekeeping gene. Quantitative real-time PCR was performed in triplicate using SYBR Green or the miScript SYBR Green PCR Kit on a Real-Time Detection System (Eppendorf).

The following mice primer sequences were used: *COL1A1*, 5′-TAAGGGTCCCCAATGGTGAGA-3′ and 5′-GGGTCCCTCGACTCCT ACAT-3′; *COL3A1*, 5′-CCTGGCTCAAATGGCTCAC-3′ and 5′-CAGGA CTGCCGTTATTCCCG-3′. Mouse miR-29 primers were purchased from Qiagen (miScript Primer assays #218300-MS00001372, MS000 05936, MS00001379).

## Isolation of mouse VSMC

Mouse VSMC were cultivated from their aortas by use of a modified procedure of Ross (Chamley-Campbell *et al*, 1979). In short, mouse thoracic aortas were removed and washed with RPMI 1640 medium. The intima and inner two-thirds of the media were carefully dissected from the vessel under an anatomic microscope, cut into pieces (1 × 1 × 0.1 mm), and planted onto a gelatin-coated (0.02%) plastic bottle (Falcon). The bottle was incubated upside down at 37°C in a humidified atmosphere of 95% air/5% $CO_2$ for 3 h, and then, medium supplemented with 20% FCS, penicillin (100 U/ml), and streptomycin (100 μg/ml) was slowly added. Cells were incubated at 37°C for 7–10 days and passaged by treatment with 0.05% trypsin/0.02% EDTA solution. Experiments were conducted on SMCs that had just achieved confluence.

## Pre-miR and anti-miR transfection

Vascular smooth muscle cells were plated at 60–70% confluency on the day before transfection. Mouse Pre-miR™ miRNA precursors were synthesized by Applied Biosystems/Ambion and Mercury™ LNA-anti-miRs by Exiqon. The following sequences were used: LNA-29b: ACTGATTTCAAATGGRGCT; LNA-30c: TGAGAGTGTAG-GATGTTTAC, LNA-CTL: GTGTAACACGTCTATACGCCCA, Pre-miR-29b: UAGCACCAUUUGAAAUCAGUGUU; Pre-miR-30c: UGUAAA-CAUCCUACACUCUCAGC; Pre-miR-CTL: sequence not specified. LNA inhibitors and precursor miRNA were transfected at a final concentration of 100 nM using Lipofectamine™ RNAiMAX (Invitrogen) according to the manufacturer's recommendations.

## Conditioned medium for proteomics analysis

Primary VSMC were carefully washed in serum-free medium to maximize the removal of proteins from the bovine serum supplement and then incubated in fresh serum-free medium for 72 h ("conditioned medium"). The conditioned medium was collected and centrifuged at 3,000 *g* for 10 min to remove cell debris. The supernatant was transferred into a new tube and stored at −80°C. A total of 2 ml of conditioned medium was first desalted using Zeba Spin desalting columns (Thermo Scientific), vacuum–dried, and resuspended in 60 μl of dd$H_2O$. A total of 30 μl was used for the proteomic analysis and 30 μl to run a SDS–PAGE for immunoblotting. Samples were denatured with 4× sample loading buffer at 96°C for 5 min and then separated in Bis-Tris discontinuous 4–12% poly-acrylamide gradient gels (NuPAGE; Invitrogen). After electrophoresis, the gels were stained using Coomassie staining and samples processed for LC/MS as described (Abonnenc *et al*, 2013). In brief, tryptic peptides were separated on a nanoflow LC system (Dionex UltiMate 3000, UK) and eluted with a 70-min gradient and eluate coupled to a nanospray source (Picoview). Spectra were collected from a high-mass accuracy analyzer (LTQ Orbitrap XL;

ThermoFisher Scientific) using full ion scan mode over the mass-to-charge (*m/z*) range 450–1,600. MS/MS was performed on the top six ions in each MS scan using the data-dependent acquisition mode with dynamic exclusion enabled. MS/MS mass spectra were extracted by extract_msn.exe version 5.0. All MS/MS samples were analyzed using Mascot (Matrix Science, London, UK; version 2.3). Mascot was searched with a fragment ion mass tolerance of 0.80 Da and a parent ion tolerance of 10.0 ppm. Iodoacetamide derivative of cysteine was specified in Mascot as a fixed modification. Peptide identifications were accepted if they could be established at > 95.0% probability as specified by the Peptide Prophet algorithm. Protein identifications were accepted if they could be established at > 99.0% probability and contained at least 2 independent peptides. Protein probabilities were assigned by the Protein Prophet algorithm. Proteins that contained similar peptides and could not be differentiated based on MS/MS analysis alone were grouped to satisfy the principles of parsimony. Mass spectrometry data are deposited in PRIDE (https://www.ebi.ac.uk/pride/archive/) under the following accession numbers: Project accession: PXD003938; Project DOI: 10.6019/PXD003938.

## MicroRNA target prediction

The prediction of miR-29b and miR-30c targets was performed with the following algorithms: TargetScan 5.2 (http://www.targetscan.org/), PicTar (http://pictar.mdc-berlin.de/), and DIANA MicroT v.4.0 (http://diana.cslab.ece.ntua.gr/).

## Immunoblot analysis of VSMC proteins

Samples were mixed with 4× denaturing sample buffer, heated at 95°C for 10 min, and separated on a Bis-Tris discontinuous 4–12% polyacrylamide gradient gel (NuPAGE; Invitrogen). Proteins were then transferred on nitrocellulose membranes. Membranes were blocked with 5% fat-free milk powder in PBS-Tween and probed for 16 h at 4°C with a primary antibody to COL3A1 (rabbit, Santa Cruz, sc-28888) at 1:500 dilution in 5% BSA. The membranes were treated with the appropriate secondary, horseradish peroxidase (HRP)-conjugated antibodies (Dako) at 1 : 2,000 dilution. Finally, the blots were imaged using enhanced chemiluminescence (ECL; GE Healthcare), and films were developed on a Xograph processor. The densitometry for the lanes from developed films was measured using the ImageJ software.

## Statistical analysis

In all studies, analyses were performed blindly by a third party, and randomization was applied when animals were injected with saline, LNA-control, or LNA-miR-29. Data are presented as mean ± the standard error of the mean (SEM) (*n* is noted in the figure legends). Groups were compared with Student's *t*-test for parametric data. When comparing multiple groups, data were analyzed by analysis of variance (ANOVA) with Bonferroni's post-test. Significance was accepted at the level of $P < 0.05$. Data analysis was performed using SigmaPlot 11.0 (Systat Software, Inc., San Jose, CA) and GraphPad Prism 6.0a software (GraphPad, San Diego, CA). We used the QSpec software proposed as a measure to detect differential expression of proteins. QSpec utilizes a hierarchical Bayes estimation of generalized linear

## The paper explained

### Problem

In spite of effective lipid-lowering therapies, atherosclerosis and plaque-associated death remains the leading cause of death worldwide. The initiation of atherosclerosis and formation of fatty streak atheroma begins within the first decade of life, resulting in substantial plaques by the fourth decade of life. While individuals can remain asymptomatic if blood can perfuse tissues downstream of the plaque, plaque rupture often occurs, resulting in thromboembolization that affects a catastrophic tissue ischemia. This signals the need for research and treatment that can induce or improve the stabilization of plaques in order to eventually affect patient mortality outcomes.

### Results

In this paper, we present data indicating that antagonism of miR-29 affects atherogenesis and plaque stability, which previously had not been explored. Chronic treatment of $ApoE^{-/-}$ mice with LNA-miR-29 markedly reduces systemic atherosclerosis and individual plaque size. LNA-miR-29 also improves fibrous cap thickness and decreases necrotic core size, both parameters consistent with more stable plaques. Specifically LNA-miR-29 treatment significantly increases ECM components in the plaque compared to adjacent non-plaque vessel but does not cause fibrosis in several tissues. Proteomic analysis of murine VSMC treated with miR-29 mimic or inhibitor demonstrated that ECM proteins are highly regulated by miR-29. Thus, antagonism of endogenous miR-29 reduces atherogenesis and improves indices of plaque instability by remodeling ECM.

### Impact

These novel data indicate that regulating miR-29 in vivo significantly lessens lesion area and beneficially affects ECM in plaques. Elucidating this newly discovered role of miR-29 regulating plaque composition will further clarify the vascular pathologies of CVD and illuminate new targets for early diagnosis and therapeutic intervention in atherosclerosis.

mixed-effects model (GLMM) to share information across the protein levels (Choi et al, 2008). This eliminates some of the assumptions needed for standard statistical tests and can increase the power of the analysis when there are a limited number of replicates available. The false discovery rate (FDR) is calculated with mixture model-based method of local FDR control based upon the Bayes factors. We considered proteins with a Bayes Factor < 10 and an FDR < 5%, including those with fold changes above 30%, to be significant.

**Expanded View** for this article is available online.

## Acknowledgements

This work was supported by grants from the National Institutes of Health (R01 HL64793, R01 HL61371, R01 HL081190, and P01 HL1070295R01 to WCS; R01 HL107953 and R01 HL106063 to CFH; and R01 HL105945 to YS) and the Fondation Leducq (MIRVAD network) to WCS, MM, TT, and CFH and an American Heart Association Postdoctoral Grant to VU. PP received support from M. Ziche at University of Siena (Italy) and an Italian Society of Pharmacology fellowship.

## Author contributions

VU and WCS designed the experiments, analyzed the data, and wrote the manuscript. VU, NR, EA, AL, PS, MA, PP, XY, AB, KLL, PZ, and BA conducted the experiments, analyzed the data, and reviewed the manuscript. RLM, TT, KM, YS, MM, and CFH analyzed the data and reviewed the manuscript.

## Conflict of interest

The authors declare that they have no conflict of interest.

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
