## [Review Process File · EMBO Molecular Medicine]

Chronic miR-29 antagonism promotes favorable plaque remodeling in atherosclerotic mice

V. Ulrich, N. Rotllan, E. Araldi, A. Luciano, P. Skroblin, M. Abonnenc, P. Perrotta, X. Yin, A. Bauer, K. Leslie, P. Zhang, B. Aryal, R.L. Montgomery, T. Thum, K. Martin, Yajaira Suarez, M. Mayr, C. Fernandez-Hernando and W. C. Sessa

Corresponding author: William C. Sessa, Yale University School of Medicine

Review timeline:

Submission date:	03 November 2015
Editorial Decision:	07 December 2015
Revision received:	06 April 2016
Editorial Decision:	29 March 2016
Revision received:	01 April 2016
Accepted:	06 April 2016

Transaction Report:

Editor: Céline Carret

1st Editorial Decision

07 December 2015

Thank you for the submission of your manuscript to EMBO Molecular Medicine. We have now heard back from the three referees whom we asked to evaluate your manuscript. Although the referees find the study to be of potential interest, they also raise a number of concerns that need to be thoroughly addressed in the next final version of your article.

As you will see from the set of comments pasted below, all referees find the topic of the study to be of interest. However, poor representation of the data, missing details, low quality images, inappropriate controls, unclear statistic usage as well as insufficient explanations often precluded appropriate understanding of the results, rendering the reviewing task difficult for the referees. Nevertheless, referees had overlapping concerns and should you be able to revise your work according to their comments, given the medical relevance of your work, we would be happy to consider your revised manuscript. However, I would like to insist that particularly in this case, the referees would need to be fully satisfied with the revision for the manuscript to go forward. For this reason, I would strongly advise against sending back an incomplete revision. Please note that EMBO Press's policy only allows a single round of major revision, and that acceptance or rejection of the manuscript will therefore depend on the completeness of your response and the satisfaction of the referees with it.

Revised manuscripts should be submitted within three months of a request for revision; they will otherwise be treated as new submissions, except under exceptional circumstances in which a short extension is obtained from the editor.

I look forward to seeing a revised form of your manuscript as soon as possible.

***** Reviewer's comments *****

Referee #2 (Comments on Novelty/Model System):

The study investigate a 14 wks treatment of miR-29 and show increase in collagen expression in plaques and in vitro experiment confirm that miR-29 could reduce protein expression of collagen I and III. This is what the paper really show, the other effects on plaque content of cells are mainly related to the increase of collagen content and are rather associative in nature.

Referee #2 (Remarks):

In this study, Ulrich and colleagues have investigated the effects of chronic inhibition of miR-29 on atherosclerotic plaque phenotype in ApoE-deficient mice. By antisense interference (LNA-miR-29), the authors achieved a systemic reduction of miR-29 expression which was reflected by concomitant reduction of atherosclerosis burden in both aortic root and brachiocephalic artery (when compared with control LNA). The authors have also found differences in plaque composition, showing the plaques of treated mice thicker fibrous cap, higher content of VSMCs and increased plaque collagen (both in mRNA expression and picrosirius red staining), features associated with a less vulnerable plaque phenotype. Gain- and loss-of-function in vitro experiments have shown the ability of mir-29 to target several collagen subtypes and other ECM-related protein (mir-29 overexpression decreases also MMP2 and fibronectin I). The authors conclude that inhibition of mir-29 might represent a new therapeutic approach for stabilizing atherosclerotic plaques.

The topic of the study is of interest, as it suggests a new potential approach to improve atherosclerotic plaque stability. The main strength of the study is the in vivo nature, allowing the authors to highlight the potential therapeutic perspective for anti-miR-29, and the in vitro confirmation of mir-29 effects on ECM components. Nevertheless, some limitations exist and could be resolved in order to improve the interpretation of the results:

1) Amongst the key findings, authors show an increase of VSMCs and fibrous cap thickness. Despite the thickening of fibrous cap could be explained by the effects on collagen synthesis, the effect of miR-29 inhibition on VSMCs content cannot be fully reconciled and explained from the current data. Is this a relative effect due to reduction of total plaque volume (or decrease in cellularity) or an absolute effect? The authors have found no significant increase in VSMCs proliferation. However,, Figure S4B revealed a trend towards an increase of Ki-67 staining in a-SMA+ cells. Is it possible that lack of significance is the result of a lack of statistical power? Could the authors provide more details on how the sample size for the experiment was estimated. Furthermore, definition of the VSMCs phenotype (i.e. synthetic vs contractile) could provide further information regarding the impact of miR-29 on this cell population. The authors could address these issues or comment on that. Finally, the method for quantification of VSMCs content (presumably % a-SMA/total plaque volume) should be included on page 13.

2) Similarly, the mechanisms behind the reduction of necrotic core is not elucidated. The authors have not found differences in proliferation and apoptosis of CD68+ cells and VSMCs, but the efferocytosis activity has not been investigated. Do the authors have data on the role of miR-29 in regulation of macrophage efferocytosis?

3) The authors show an increase of collagen I and III expression in atherosclerotic plaques, but not an increase in fibrosis in heart, lung, liver and spleen. Due to the relevance of the absence of aberrant fibrosis after LNA-miR-29 treatment, a quantification for the images shown in Suppl. Figure 5 should be provided.

4) Control groups: the authors have used 2 different controls for their experiments: saline and a negative LNA treatment. Remarkably, the significant results shown in figure 1C, 2E, 3B are obtained mainly after comparison with the "LNA-control". Moreover, treatment with LNA-control resulted in changes in lipid profile (i.e. Suppl. Fig 1A: increase in LDL and decrease in HDL

cholesterol). Is it possible that LNA-control is actually targeting an anti-atherosclerotic target?
COuld the authors possibly test a different negative LNA?

5) In vitro experiments: the authors performed gain- and loss-of-function experiments on VSMCs in order to confirm the effects on protein targets. Interestingly, Fig. 3C show that LNA-miR-29 increase collagen 3 levels when compared with LNA-control. Again, apparently LNA-control results in an increase in collagen 3 levels (with a trend of increase also toward the other "negative control" miR-30). Would it be possible for the authors to test and include a different negative control? Furthermore, miR-30 used as further negative control has been involved in regulation of TGF-beta pathway as well, thus potentially affecting fibrosis development per se (e.g. Roy S et al. 2015, Tu X et al. 2015, Duisters et al 2009).

6) Supplementary Figure 3: although the quantification and the title of the figure caption state that LNA-miR-29 does not affect macrophage content in atherosclerotic plaques, the representatives figures apparently suggest a rather strong effect. While this could reflect a substantial variability in the effect, the authors could perhaps consider revise this figure with a figure more representative of the general trend.

7) Statistics: the methods used for verifying the assumption of the test used for rejecting null-hypothesis (e.g. in case of ANOVA) should be explained.

Referee #3 (Remarks):

Summary:

In their manuscript "Chronic miR-29 antagonism promotes favorable plaque remodeling in atherosclerotic mice", Ulrich and colleagues study the impact of miR-29 expression modulation on atheroma formation and plaque composition. They report that chronic administration of LNA-miR-29 into an atherosclerotic mouse model increases the expression levels of col1A and col3A, which are key elements of the plaque extracellular matrix (ECM). The author's claim that the ECM composition changes decrease atherosclerotic lesion size, enhance fibrous cap thickness and reduce plaque necrosis. They further show that LNA-miR-29 administration does not affect circulating lipids, blood chemistry or ECM of other solid organs. It is concluded that LNA-miR-29 treatment might be a novel therapeutic strategy for atherosclerotic vessel disease that is independent of lipid-lowering activities.

Novelty:

Use of LNA-miR-29 blocking oligonucleotides as a novel therapeutic approach for the treatment of atherosclerosis.

General comment:

This manuscript studies how pharmacological inhibition of the miR-29 family affects atherosclerosis in mice. A combination of both in-vivo and in-vitro experiments is used to show convincingly that blocking the biological activity of miR-29 with LNA-based oligos decreases atherosclerotic lesion formation. Overall, this manuscript is well written, the data are solid, and the results consistent with the well-established role of the miR-29 family in regulating components of the ECM. The findings are not strikingly novel, nevertheless relevant and interesting from a medical perspective.

I enclose some comments the authors may want to address to further the study.

Specific comments - major:

Figure 4 (A) - To be able to judge the relevance of ECM regulation by miR-29, it would be important to see a more comprehensive list of the regulated proteins from the proteome data set.

Figure 4 (A) -The authors claim that fibronectin 1 (FBN1) and MMP2 are miR-29 targets. Is the expression of these two proteins up-regulated when the anti-miR-29 is expressed? The authors should include luciferase reporter assays to validate the interaction between miR-29, FBN1 and MMP2.

Material and Methods - This section lacks information how miR-29a, -29b and -29c expression levels were assessed by qPCR (Supplementary Figure 1). The authors should provide detailed information in the Material and Methods section.

Supplementary Figure 1 (A) - The authors claim that administration of either LNA-control or LNA-miR-29 had no effect on total cholesterol levels. As illustrated, it seems, however, that there is a significant difference in total cholesterol levels when compared to the saline control. What is the explanation for this observation? In a recent paper (Kurtz et al., Scientific Reports 2015), it was also reported that in vivo blockade of miR-29 by LNA-based oligonucleotides reduces plasma cholesterol levels by 40%. How do the authors explain these different results?

Supplementary Figure 1 (B) - The expression levels of the miR-29 family members were quantified in several mouse tissues after administration of LNA-control and LNA-miR-29. However, the experimental set-up used ("10mg/kg twice in the first week then weekly for a month") is different from the one depicted in Figure 1 ("4mg/Kg biweekly for 14 weeks"). What is the rationale for the different treatment regimens?

Supplementary Figure 1 (B) - Which housekeeping gene was used to normalize the qPCR data? Such information is absent in the Material and Methods section. The Y-axis labels should be replaced since a microRNA is not a messenger RNA (mRNA).

Specific comments - minor:

Title- Typographical error: "antagonism" should be replaced by "antagonism".

Discussion- I do not understand the of the meaning of "The mechanistic basis for why mimicry or antagonism of miRNAs does not affect physiological processes is not known." Please explain or rephrase.

Material and Methods -"Experimental Animals" - The authors provide a short description on the pharmacokinetic profiles of LNA-based oligonucleotides. This information is not particularly helpful in the material and methods section. Such information should be integrated somewhere else in the manuscript.

Referee #4 (Remarks):

The manuscript reports improved plaque morphology associated with chronic administration of LNA-miR-29. Overall the manuscript is well written and concise. However, the some differences between the control and LNA-miR-29 cohorts are hard to appreciate on the included images. I think the manuscript could be improved by addressing the following comments.

- 1.) The first paragraph of the results section and the methods both cite a LNA dose of 4mg/kg. However the figure legend of Supp. Fig. 1 cites a different dosing schedule and a dose of 10mg/kg.
- 2.) LNA-miR29 appears to inhibit miR-29a,b,c equally per Supp. Fig. 1. Is it a pool of isoform specific oligos or a single sequence that tolerates some mismatches?
- 3.) Supp. Fig 1 shows no significant reduction in miR-29 levels in skin yet the methods section describes "...significant oligo accumulation in spleen, bone marrow and distal skin." Also if there is significant oligo accumulation in bone marrow and spleen is miR-29 repressed in macrophages and/or other leukocytes that may impact atherosclerotic lesion development and stability?
- 4.) Figure 3a shows increased COL1A1 and COL3A1 expression in plaque vs non-plaque tissue. Was miR-29 equally repressed in both plaque and non-plaque tissues?
- 5.) Why was miR-30 used as the control in Figure 4?
- 6.) In Figure 1, are the lesions in LNA-miR-29 mice significantly different from saline control mice or only LNA-control mice? The difference in sizes is not very large at the selected time point on this diet. Have other time points and/or diets been tested?
- 7.) Figure 2 uses two panels of pentachrome stained sections to shows fibrous cap and necrotic areas. Why couldn't necrosis and fibrous cap be shown on the same image? This would allow individual panels to be larger and therefore may more readily highlight the differences between the groups (particularly for BC fibrous cap area). It may also be helpful to outline the total plaque for

reference. Also no black bar is shown in 2A root saline.

8.) The title line of the legend for Supp. Fig 1 was incomplete on my copy.

9.) At first glance there appears to be significantly less CD68 (green) staining in Supp. Fig 3A LNA-miR-29 (upper left panel). Perhaps a different magnification or better delineation of the areas quantified in Supp. Fig. 3B would better illustrate the relevant comparisons.

10.) It is unclear what the magnified inset boxes are intended to show in Supp. Fig. 4A as, for example, the boxes for saline and LNA-control both show substantially more Ki-67 (green) staining than LNA-miR-29 yet the graph in Supp. Fig. 4B shows the opposite trend.

1st Revision - authors' response

06 April 2016

EMBO MM-miR-29 paper

We would like to thank the reviewers for their insightful reviews. Please find the original comments followed by our rebuttals in bold font.

Rev 2.

1) Amongst the key findings, authors show an increase of VSMCs and fibrous cap thickness. Despite the thickening of fibrous cap could be explained by the effects on collagen synthesis, the effect of miR-29 inhibition on VSMCs content cannot be fully reconciled and explained from the current data. Is this a relative effect due to reduction of total plaque volume (or decrease in cellularity) or an absolute effect?

This is a good question which we addressed by evaluating total DAPI stained nuclei in the plaque area. As seen below the total cellularity does not change, (p=0.252 by One-way ANOVA, n=4, 5, 5 mice/group) thus we conclude that the effect of miR-29 on plaque size is a reduction in total volume.

1A. The authors have found no significant increase in VSMCs proliferation. However, Figure S4B revealed a trend towards an increase of Ki-67 staining in α -SMA+ cells. Is it possible that lack of significance is the result of a lack of statistical power? Could the authors provide more details on how the sample size for the experiment was estimated.

While this is a valid question and one we have considered, we have utilized stringent statistical analyses and do not believe this trend would become significant with a higher n, due to the interrogatory quantification methods used herein, which were performed in the manner as previously published. [Rotllan et al. (2015) Genetic evidence supports a major role for Akt1 in VSMCs during atherogenesis. *Circulation* 116, 1744-1762.] Briefly, slide sample size for this quantification was such that we analyzed 3 randomly selected but distinctly separate sections from the aortic root plaques of each mouse, then averaged these values per mouse so that n=9, 13, 15 per group on the graph. To explain further, 30 consecutive cryosections were sectioned inferiorly from the ascending aorta down through the heart for each mouse. Sections were collected from each mouse's root beginning when the plaque was first detected, and slides were numbered #1-30 in order of collection, so slide #1 would be closest to the ascending aorta and slide #30 was closest to the apex. For each mouse three slides were chosen at depth

intervals labeled #2, 14, an 28, and stained for SMA and Ki-67⁺. Then co-localization of the two stainings was quantified using ImageJ software for each slide, and the three slides from each mouse were averaged independently, then as a group, such that for each group there were 27, 39, and 45 slides analyzed and 9, 13, and 15 averages of Ki-67⁺ and SMA⁺ cells counted and graphed. We used ANOVA and the stringent Bonferroni post-hoc test to statistically compare these averages in each group. Mouse sample size was chosen such that we started with an average of 4 saline, and 6 LNA-control and 6 LNA-miR-29 in each group for each of the three independent cohorts.

1B. Furthermore, definition of the VSCMs phenotype (i.e. synthetic vs contractile) could provide further information regarding the impact of miR-29 on this cell population. The authors could address comment on that.

This is a great question which we considered initially, but due to the standard harvest timepoint we did not believe we could make a fair assessment of the transitioning phenotype of the VMSCs in plaques at this one timepoint.

1C. Finally, the method for quantification of VSMCs content (presumably % α -SMA/total plaque volume) should be included on page 13.

The method of quantification was calculated as SMA staining intensity/percent of total plaque volume.

2) Similarly, the mechanisms behind the reduction of necrotic core is not elucidated. The authors have not found differences in proliferation and apoptosis of CD68⁺ cells and VSMCs, but the efferocytosis activity has not been investigated. Do the authors have data on the role of miR-29 in regulation of macrophage efferocytosis?

This is a great question that we have addressed below in preliminary experiments included below. We modified an online protocol from the Tabas laboratory (www.tabaslab.com/protocols), prepared by Nadolski and Thorp, 2009. Utilizing RAW macrophages and apoptotic Jurkat cells, we evaluated % of efferocytosis (as published by Wan...Thorp et al., *Circ Res*, 2013) at several time points. Briefly RAW cells were treated with 100nM LNA-control or LNA-miR-29, and after 48 hours, exposed to apoptotic Jurkat cells. Of note, RAW were plated and Jurkat incubated at equal number in both treatment groups, and RAW cell numbers did not vary significantly during the course of the experiment. Furthermore, miR-29 levels in the RAW cells were decreased by over 98%. We have conducted this experiment two times (see individual experiments below), and do not at this point see a difference between the LNA-Control and LNA-miR-29 treated groups. Thus, short term LNA-miR-29 treatment does not influence macrophages efferocytosis, however, the role of miR-29 in regulating efferocytosis during lesion formation is yet to be determined.

3) The authors show an increase of collagen I and III expression in atherosclerotic plaques, but not an increase in fibrosis in heart, lung, liver and spleen. Due to the relevance of the absence of aberrant fibrosis after LNA-miR-29 treatment, a quantification for the images shown in Suppl. Figure 5 should be provided.

This was noted and provided. When quantified there was no significance among the groups. These data corroborate the results from Boon et al., *Circ Res*, 2011 (reference #24), in which five times the LNA-miR-29 dose was utilized in an aneurysm and aging model with no fibrosis evident in the above target tissues.

4A) Control groups: the authors have used 2 different controls for their experiments: saline and a negative LNA treatment. Remarkably, the significant results shown in figure 1C, 2E, 3B are obtained mainly after comparison with the "LNA-control". Moreover, treatment with LNA-control resulted in changes in lipid profile (i.e. Suppl. Fig 1A: increase in LDL and decrease in HDL cholesterol). Is it possible that LNA-control is actually targeting an anti-atherosclerotic target?

To evaluate endpoints we utilized saline vehicle and LNA-Control negative controls to ensure that the LNA-control did not significantly affect measurements when compared with the vehicle group. These 2 control were used because there is evidence at certain doses LNA modification of oligos can cause hepatotoxicity [Swayze et al. (2007) Antisense oligonucleotides containing locked nucleic acid improve potency but cause significant hepatotoxicity in animals. *Nucleic Acids Research* 35]. Since we had three treatment groups, we chose to use one way ANOVA with a stringent Bonferroni post test to analyze the groups. Only graphs marked with an asterisk are significant with a p value less than 0.05.

4B. Could the authors possibly test a different negative LNA?

There were no significant differences in lipoprotein pools between the groups, and as expected in ApoE^{-/-} mice, the IDL/LDL peak is elevated at the expense of the HDL fraction. When comparing circulating immune cells via flow cytometry, the LNA-control group does not differ significantly from the saline group. Arguably the two cell types most crucial to atherosclerotic progression are not affected by LNA-control treatment (compared to saline vehicle) in terms of proliferation or apoptosis (supplementary figure 4). Finally, the LNA-control is a *c. elegans* sequence that was designed such that it would not target mammalian microRNA, therefore it is doubtful that the LNA-control targets an anti-atherosclerotic target.

5A) In vitro experiments: the authors performed gain- and loss-of-function experiments on VSMCs in order to confirm the effects on protein targets. Interestingly, Fig. 3C show that LNA-miR-29 increase collagen 3 levels when compared with LNA-control. Again, apparently LNA-control results in an increase in collagen 3 levels (with a trend of increase also toward the other "negative control" miR-30). Would it be possible for the authors to test and include a different negative control?

In the gain and loss of function experiments in primary VSMCs in Figure 4C, the reagents used are not LNAs, but are classic antimirRs and mimics of miR-29 and miR-30. These treatments were chosen as these parameters were previously published by Abonnenc et al., *Circ Res*, 2013 (reference #11). Furthermore, while miR-29 and miR-30 are both expressed in VSMCs, *Col3A1* is not a predicted or confirmed target of miR-30 (Targetscan/miRBase). miR 30 was just an additional control!

5B. Furthermore, miR-30 used as further negative control has been involved in regulation of TGF-beta pathway as well, thus potentially affecting fibrosis development per se (e.g. Roy S et al. 2015, Tu X et al. 2015, Duisters et al 2009).

While we acknowledge the work done in the papers cited by the reviewer, there are several notable differences between our study and those in the papers provided that lend to our decision to use miR-30 as a control. First, these papers evaluated the role of miR-30 in TGF- b induced cardiac/liver fibrosis while in our study we did not treat cells with TGF- b. Instead we chose to evaluate how miR-29 manipulation changes the secretome without mimicking fibrosis, since we did not see fibrosis in our LNA-miR-29 mice. Furthermore, Roy et al. showed that miR-30 decreased *Col1A1* and *Col5A3* levels, but these genes are not validated

targets of miR-30 and *Col3A1*, the most differentially induced confirmed target of miR-29 in our study, is not a predicted or confirmed target of miR-30. Indeed, both miR-29 and 30 could regulate fibrosis, however the pathways by which each miR effects these pathologies is difference such that miR-29 targets ECM synthesis and miR-30 targets CTGF signaling (Pandit et al., 2010, *Am J Respir Crit Care Med*, Mann et al., 2010, *Gastroenterology*). Technically speaking, the cell types used were very different in the quoted studies (TGF-treated hepatocytes (Roy and Tu), and cardio- myocytes/fibroblasts (Duisters)) compared to our study in quiescent VSMCs.

6) *Supplementary Figure 3: although the quantification and the title of the figure caption state that LNA-miR-29 does not affect macrophage content in atherosclerotic plaques, the representative figures apparently suggest a rather strong effect. While this could reflect a substantial variability in the effect, the authors could perhaps consider revise this figure with a figure more representative of the general trend.*

Thank you. This figure has been amended.

7) *Statistics: the methods used for verifying the assumption of the test used for rejecting null-hypothesis (e.g. in case of ANOVA) should be explained.*

Our original null hypothesis was that decreasing miR-29 levels would not affect indices of plaque stability, and that each of the group means would be equal. Using three groups of different animals with different treatments, our samples were appropriately independent to utilize a one way ANOVA and a rigorous post-hoc test. This was confirmed by a statistician at Yale (Dr. Hongyu Zhao) prior to experimentation. We appreciate that the reviewer is most likely asking about the normality assumption and homogeneity of variances assumption in ANOVA. The former assumption can be visually checked by Q-Q plot or tested by Shapiro-Wilk test on the variables or residues. The homogeneity of variance be checked by Bartlett or Levene's test.

Comments to Reviewer 3.

Specific comments - major:

1. *Figure 4 (A) - To be able to judge the relevance of ECM regulation by miR-29, it would be important to see a more comprehensive list of the regulated proteins from the proteome data set.*

Supplementary figure 6 addresses this requirement. Proteins in the table have an FDR<0.05.

2. *Figure 4 (A) -The authors claim that fibronectin 1 (FBN1) and MMP2 are miR-29 targets. Is the expression of these two proteins up-regulated when the anti-miR-29 is expressed? The authors should include luciferase reporter assays to validate the interaction between miR-29, FBN1 and MMP2.*

Previous validation work for FBN1 (Segupta et al., 2008, pubmed ID#18390668 and van Rooij, 2008, pubmed ID#18723672) and MMP2 (Liu et al., 2010, pubmed ID#20194304) have already been confirmed targets of miR-29 using mRNA and luciferase assays. In our secretome study, FBN1 secretion was significantly affected in the anti-miR-29 treated cells (supp.fig. 6), and MMP2 was secreted but not significant (Bayes Factor=0.895, Fold change=-0.124, FDR=1). The effect of anti-miR on the magnitude of derepression is due to the abundance of the endogenous miR and its loading of a specific mRNA into the RISC complex.

3. *Material and Methods - This section lacks information how miR-29a, -29b and -29c expression levels were assessed by qPCR (Supplementary Figure 1). The authors should provide detailed information in the Material and Methods section.*

This concern is noted and details were added.

4. *Supplementary Figure 1 (A) - The authors claim that administration of either LNA-control or LNA-miR-29 had no effect on total cholesterol levels. As illustrated, it seems, however, that there is a significant difference in total cholesterol levels when compared to the saline control. What is the explanation for this observation?*

Statistically there was no difference, and this effect was confirmed in independent cohorts of animals.

5. *In a recent paper (Kurtz et al., Scientific Reports 2015), it was also reported that in vivo blockade of miR-29 by LNA-based oligonucleotides reduces plasma cholesterol levels by 40%. How do the authors explain these different results?*

We have seen and appreciate the findings in this paper, but note several key differences with our studies. First, Kurtz et al. studied C57BL/6J WT female mice on a chow diet, while we utilized C57BL/6J *ApoE*^{-/-} male mice on an atherogenic diet for our analysis. The lipogenesis and the resulting lipoprotein profile for these mice would be very different, with HDL being the largest fraction in WT mice, and VLDL being the largest fraction with almost undetectable HDL in an *ApoE*^{-/-} mouse. With total cholesterol being a log higher in our mice, these lipid differences would effect a different inflammatory response and pathology. Furthermore, the Kurtz paper used five times the dose of LNA that we utilized. Most importantly, the end points were different and time to harvest very different, as we were interested in long term (14 weeks) disease pathology compared to short term (1 week) gene effects in presumably healthy mice. While we have not personally used the Exicon LNAs against miR-29, Kurtz did not use an LNA control in their study and wonder if there is liver inflammation [Swayze et al. (2007) Antisense oligonucleotides containing locked nucleic acid improve potency but cause significant hepatotoxicity in animals. *Nucleic Acids Research* 35].

6. *Supplementary Figure 1 (B) - The expression levels of the miR-29 family members were quantified in several mouse tissues after administration of LNA-control and LNA-miR-29. However, the experimental set-up used ("10mg/kg twice in the first week then weekly for a month") is different from the one depicted in Figure 1 ("4mg/Kg biweekly for 14 weeks"). What is the rationale for the different treatment regimens?*

The characterization of the LNA was at 10mg/kg, however we have repeated these studies using the 4mg/kg dose, and thus have included to this data which is more appropriate and relevant to our study.

7. *Supplementary Figure 1 (B) - Which housekeeping gene was used to normalize the qPCR data? Such information is absent in the Material and Methods section. The Y-axis labels should be replaced since a microRNA is not a messenger RNA (mRNA).*

These data are normalized to RNU6. Since there was concerned about the higher dose of LNA, we have repeated the experiment at the dose amount used throughout the paper (4mg/kg). In this new panel (supp. fig 1B) we have confirmed targeting of the LNA by highlighting significantly reduced levels of miR-29a/b/c in vessels.

Specific comments - minor:

8. *Title- Typographical error: "antagonism" should be replaced by "antagonism".*

This has been corrected.

9. *Discussion- I do not understand the meaning of "The mechanistic basis for why mimicry or antagonism of miRNAs does not affect physiological processes is not known." Please explain or rephrase.*

This has been amended to "The mechanistic basis for why LNA-miR-29 only affects diseased tissue is not known."

10. *Material and Methods - "Experimental Animals" - The authors provide a short description on the pharmacokinetic profiles of LNA-based oligonucleotides. This information is not particularly*

helpful in the material and methods section. Such information should be integrated somewhere else in the manuscript.

This has been amended and moved into the results section.

Comments to Referee #4

1.) The first paragraph of the results section and the methods both cite a LNA dose of 4mg/kg. However the figure legend of Supp. Fig. 1 cites a different dosing schedule and a dose of 10mg/kg.

This concern has been noted by another reviewer, and we have amended this panel to include new data from mice treated with 4mg/kg documenting targeting in tissues (supp. fig. 1).

2.) LNA-miR29 appears to inhibit miR-29a,b,c equally per Supp. Fig. 1. Is it a pool of isoform specific oligos or a single sequence that tolerates some mismatches?

While the specifics of the LNA are largely proprietary, our collaborator has informed us that the LNA is a single sequence that tolerates some mismatches and targets all three miRs.

3.) Supp. Fig 1 shows no significant reduction in miR-29 levels in skin yet the methods section describes "...significant oligo accumulation in spleen, bone marrow and distal skin." Also if there is significant oligo accumulation in bone marrow and spleen is miR-29 repressed in macrophages and/or other leukocytes that may impact atherosclerotic lesion development and stability?

This is a very good question which we are addressing in ongoing studies, as we know that miR-29 is expressed in macrophages and T-cells. Several publications implicate miR-29 in the differentiation of and influence on IFN-g production in T-cells, as well as affecting the infiltration of macrophages in pulmonary fibrosis (Steiner et al., *Immunity*, 2011, 35, Smith et al., *J Immunol* 2012, 189, Ma et al., *Nature Immunology* 2011, 12, Xiao et al., *Molecular Therapy* 2012, 20). While we do not see changes in circulating levels or in plaques between our study groups, we cannot exclude that regulating miR-29 might influence the activity or differentiations of these cells in the early inflammatory stages during initiation of fatty streak atheromas. Therefore bone marrow transplant studies are being conducted to evaluate the influence of miR-29 knockout bone marrow derived monocytes and leukocytes in atherosclerosis progression and plaque remodeling.

4.) Figure 3a shows increased COL1A1 and COL3A1 expression in plaque vs non-plaque tissue. Was miR-29 equally repressed in both plaque and non-plaque tissues?

In the LNA-miR-29 treated mice, miR-29 levels were reduced to less than 5% of endogenous levels in both nonplaque and plaque tissues.

5.) Why was miR-30 used as the control in Figure 4?

miR-30 and miR-29 are both expressed in VSMC, however miR-30 does not target Col3A1, which would make the use of miR-30 as a control a valid miR choice. Furthermore, this assay was already published in which miR-30 and miR-29 were manipulated by anti-miR and mimic interventions, so most experimental conditions were already optimized (Abonnenc et al., *Circ Res*, 2013, reference #11).

6A.) In Figure 1, are the lesions in LNA-miR-29 mice significantly different from saline control mice or only LNA-control mice?

When ANOVA with Bonferroni was performed on all three groups, there is statistical significance (root p=0.002/BC p=0.003).

6B.) *The difference in sizes is not very large at the selected time point on this diet. Have other time points and/or diets been tested?*

Our original hypothesis was that lowering miR-29 levels would affect plaque composition, not lesion size, so reduced lesion size was not anticipated. We have not tested other diets or timepoints.

7.) *Figure 2 uses two panels of pentachrome stained sections to show fibrous cap and necrotic areas. Why couldn't necrosis and fibrous cap be shown on the same image? This would allow individual panels to be larger and therefore may more readily highlight the differences between the groups (particularly for BC fibrous cap area). It may also be helpful to outline the total plaque for reference. Also no black bar is shown in 2A root saline.*

We chose to separate the panels so the reader could appreciate the intricate differences highlighted by the Movat pentachrome staining in the representative images per readout. More importantly, in saline and LNA-control images the large necrotic cores encroach into the fibrous cap area, which would make it hard to visibly delineate the difference characteristics. We have amended Figure 2A to add a black bar.

8.) *The title line of the legend for Supp. Fig 1 was incomplete on my copy.*

This has been amended.

9.) *At first glance there appears to be significantly less CD68 (green) staining in Supp. Fig 3A LNA-miR-29 (upper left panel). Perhaps a different magnification or better delineation of the areas quantified in Supp. Fig. 3B would better illustrate the relevant comparisons.*

This panel has been amended such that the representative image better represents the group.

10.) *It is unclear what the magnified inset boxes are intended to show in Supp. Fig. 4A as, for example, the boxes for saline and LNA-control both show substantially more Ki-67 (green) staining than LNA-miR-29 yet the graph in Supp. Fig. 4B shows the opposite trend.*

The insets show representative areas of costaining from each figure, but are not meant to be representative in the amount as is graphed in sup. fig. 4B. With that being said, the representative picture and legend have been amended to clarify this purpose.

2nd Editorial Decision

29 March 2016

Thank you for the submission of your revised manuscript to EMBO Molecular Medicine. We have now received the enclosed reports from the referees that were asked to re-assess it. As you will see the reviewers are now globally supportive and I am pleased to inform you that we will be able to accept your manuscript pending the following final amendments:

1) Please address the minor comments from referee 3. Please provide a letter INCLUDING the reviewer's reports and your detailed responses to their comments (as Word file).

Please submit your revised manuscript within two weeks and I look forward to seeing it.

***** Reviewer's comments *****

Referee #2 (Remarks):

The manuscript has been suitably and thoroughly revised and is now acceptable for publication.

Referee #3 (Remarks):

The authors have addressed my concerns in the revised version of the manuscript. The paper will be of interest to the broad readership of EMBO Molecular Medicine.

Referee #4 (Remarks):

The manuscript reports improved plaque morphology associated with chronic administration of LNA-miR-29. Overall the manuscript is well written and concise. I think the manuscript would be improved by addressing the following.

1.) The authors state that "LNA-miR-29 effectively neutralized all miR-29 isoforms in several mouse tissues (Fig. EV1B)". " but only show data for arteries. The impact on miR-29 levels in additional tissues should be shown, particularly as they note oligo accumulation is highest in the kidney and liver. While inhibition of miR-29 may not have an immediate impact on fibrosis of healthy tissues (as shown in Figure EV5), it would almost certainly cause an exaggerated response to pro-fibrotic stimuli/comorbidities. I also think showing the level of inhibition in other tissues would be of interest to readers who might be considering LNA based approaches for other conditions.

2.) The y-axis label in Fig. 3A includes "(p/np ratio)" but the graphs show individual values for both P and NP. The figure legend states "Three assays were carried out, ..., n=4,3,4." I assume n=4,3,4 respectively represents the number of mice for the saline, LNA-control, and LNA-miR-29 cohorts but it is unclear if three assays refers to three independent P and NP areas from each mouse or simply triplicate PCR amplification.

3.) The last paragraph of the results section describes COL3A1 as the most highly regulated protein following both mimic and anti-miR treatment. However, the data in Supp Fig 6 seems inconsistent with that statement as there are several proteins with larger fold changes, equivalent FDR and, at least for the mimic-29 treated, larger Bayes factors. I also think generating comparisons that show a positive fold change for well established miR-29 targets (collagen, etc) in cells treated with anti-miR-29 (panel A) as well as ranking by fold change (instead of FDR) would offer a more intuitive order for the data.

2nd Revision - authors' response

01 April 2016

Response to Reviewer #4 acceptance comments.

We would like to thank the reviewer for his/her suggestions. The comments are listed followed by our response in bold.

Referee #4 (Remarks):

The manuscript reports improved plaque morphology associated with chronic administration of LNA-miR-29. Overall the manuscript is well written and concise. I think the manuscript would be improved by addressing the following.

1.) The authors state that "LNA-miR-29 effectively neutralized all miR-29 isoforms in several mouse tissues (Fig. EV1B)". " but only show data for arteries. The impact on miR-29 levels in additional tissues should be shown, particularly as they note oligo accumulation is highest in the kidney and liver. While inhibition of miR-29 may not have an immediate impact on fibrosis of healthy tissues (as shown in Figure EV5), it would almost certainly cause an exaggerated response to pro-fibrotic

stimuli/comorbidities. I also think showing the level of inhibition in other tissues would be of interest to readers who might be considering LNA based approaches for other conditions.

This is a good point but we have not quantified miR-29 in tissues other than the vessels and do not have access to remnant tissues to complete this experiment at this stage. In pilot studies in normal mice, LNA-miR-29 did suppress miR-29 isoforms in heart, lung, and aorta when dosed at 10mg/kg, which is higher than the 4 mg/kg chronic dosing in the atherosclerosis study. We do not think we should include the 10mg/kg data since we did not use this amount in any of the in vivo data presented.

2.) The y-axis label in Fig. 3A includes "(p/np ratio)" but the graphs show individual values for both P and NP.

This normalization has been clarified in the legend in the revised MS.

The figure legend states "Three assays were carried out, ..., n=4,3,4." I assume n=4,3,4 respectively represents the number of mice for the saline, LNA-control, and LNA-miR-29 cohorts but it is unclear if three assays refers to three independent P and NP areas from each mouse or simply triplicate PCR amplification.

This is corrected in the revised MS.

3.) The last paragraph of the results section describes COL3A1 as the most highly regulated protein following both mimic and anti-miR treatment. However, the data in Supp Fig 6 seems inconsistent with that statement as there are several proteins with larger fold changes, equivalent FDR and, at least for the mimic-29 treated, larger Bayes factors. I also think generating comparisons that show a positive fold change for well established miR-29 targets (collagen, etc) in cells treated with anti-miR-29 (panel A) as well as ranking by fold change (instead of FDR) would offer a more intuitive order for the data.

This is a good point. However, while there are other proteins that were detected with larger fold changes in either the pre-miR or anti-miR treated supernatants, COL3A1 is the protein that appears higher on both sides relative to other proteins.

Corresponding Author Name: William C. Sessa

Manuscript Number: EMM-2015-06031